# miR-302 Suppresses the Proliferation, Migration, and Invasion of Breast Cancer Cells by Downregulating ATAD2

**DOI:** 10.3390/cancers14184345

**Published:** 2022-09-06

**Authors:** Yo Sep Hwang, Eun Sun Park, Byung Moo Oh, Tae Gi Uhm, Suk Ran Yoon, Jong-Lyul Park, Hee Jun Cho, Hee Gu Lee

**Affiliations:** 1Immunotherapy Research Center, Korea Research Institute of Bioscience and Biotechnology, Yuseong-gu, Daejeon 34141, Korea; 2Department of Biomolecular Science, KRIBB School of Bioscience, Korea University of Science and Technology (UST), Yuseong-gu, Daejeon 34113, Korea; 3Plumbline Life Science Inc., College of Medicine, The Catholic University of Korea, Seoul 06591, Korea; 4Personalized Genomic Medicine Research Center, Korea Research Institute of Bioscience and Biotechnology, Yuseong-gu, Daejeon 34141, Korea

**Keywords:** ATAD2, miR-302, breast cancer, proliferation, invasion, tumor growth

## Abstract

**Simple Summary:**

ATPase family AAA domain-containing protein 2 (ATAD2) overexpression is associated with poor survival and disease recurrence in multiple cancers. The current study aimed to investigate the expression and function of ATAD2 in breast cancer. Our results showed that ATAD2 expression was upregulated in human breast cancer tissues and cell lines, while ATAD2 knockdown inhibited the proliferation, migration, and invasion of breast cancer cells. Moreover, we provide evidence suggesting that miR-302 directly targets ATAD2 and thus modulates cancer cell proliferation, migration, and invasion in vitro. Moreover, ATAD2 overexpression rescued the inhibition of tumor growth caused by miR-302 in xenograft mice. These findings indicate that miR-302 plays a crucial role in inhibiting the malignant phenotypes of breast cancer cells by targeting ATAD2.

**Abstract:**

Breast cancer is the most common malignant tumor in women. The ATPase family AAA domain-containing protein 2 (ATAD2) contains an ATPase domain and a bromodomain, and is abnormally expressed in various human cancers, including breast cancer. However, the molecular mechanisms underlying the regulation of ATAD2 expression in breast cancer remain unclear. This study aimed to investigate the expression and function of ATAD2 in breast cancer. We found that ATAD2 was highly expressed in human breast cancer tissues and cell lines. ATAD2 depletion via RNA interference inhibited the proliferation, migration, and invasive ability of the SKBR3 and T47D breast cancer cell lines. Furthermore, Western blot analysis and luciferase assay results revealed that ATAD2 is a putative target of miR-302. Transfection with miR-302 mimics markedly reduced cell migration and invasion. These inhibitory effects of miR-302 were restored by ATAD2 overexpression. Moreover, miR-302 overexpression in SKBR3 and T47D cells suppressed tumor growth in the xenograft mouse model. However, ATAD2 overexpression rescued the decreased tumor growth seen after miR-302 overexpression. Our findings indicate that miR-302 plays a prominent role in inhibiting the cancer cell behavior associated with tumor progression by targeting ATAD2, and could thus be a valuable target for breast cancer therapy.

## 1. Introduction

ATPase family AAA domain-containing protein 2 (ATAD2) is a member of the AAA ATPase family that contains an AAA ATPase domain and a bromodomain [1,2]. The AAA domain is responsible for ATP binding and protein multimerization, while the bromodomain is highly conserved and mediates binding with acetylated lysine in histones and non-histone proteins [3]. ATAD2 is involved in transcriptional regulation via the modulation of chromatin remodeling and histone acetylation. It also regulates the cell cycle, differentiation, and other biological activities by modifying non-histone acetylation [4,5]. ATAD2 functions as a transcriptional co-regulator of several oncogenic transcription factors, including estrogen receptor (ER) and androgen receptor (AR) [6,7]. ATAD2 further coordinates the E2F and Myc oncogenic transcription factors [8,9]. ATAD2 is overexpressed in various human cancers, including colorectal, gastric, lung, and breast cancers [10,11,12,13,14,15]. ATAD2 overexpression is associated with poor survival and disease recurrence in multiple cancers [3], and its elevated expression is associated with chemoresistant phenotypes of the colon, gastric, and breast cancers [16,17,18]. Accordingly, ATAD2 has been proposed as a poor prognostic marker; however, the mechanisms underlying its action remain unclear.

Non-coding RNAs (ncRNAs) include a diverse range of RNA species, including microRNAs (miRNAs) and long noncoding RNAs (lncRNAs) [19]. miRNAs are small single stranded non-coding RNAs containing approximately 19–24 nucleotides. miRNAs are transcribed from DNA sequences into primary miRNAs (pri-miRNAs) and processed into precursor miRNAs (pre-miRNAs). The pre-miRNAs cross the nuclear membrane into the cytoplasm, and are further processed into mature single-stranded miRNAs by Dicer ribonuclease. Mature miRNAs adhere to the RNA-induced silencing complex (RISC), which is an effector molecule comprising miRNAs and specific proteins [20,21,22]. miRNAs modulate gene expression by targeting the 3′-untranslated region (UTR) of target mRNAs [23]. miRNAs regulate various cellular processes, including proliferation, migration, invasion, and apoptosis, by modulating the expression of target genes [24]. Accumulating evidence has suggested that abnormal miRNA expression is closely associated with carcinogenesis and the malignant progression of various cancers, including breast cancers, by suppressing the expression of oncogenes and tumor suppressor genes [25].

In the current study, we describe novel molecular mechanisms that define a new role of ATAD2 in breast cancer progression. Our results show that ATAD2 expression is upregulated in human breast cancer tissues and cell lines, while ATAD2 knockdown inhibits the proliferation, migration, and invasion of breast cancer cells. Moreover, we provide evidence suggesting that miR-302 plays a crucial role in inhibiting cancer cell proliferation, migration, and invasion in vitro and tumor growth in vivo by targeting ATAD2.

## 2. Materials and Methods

### 2.1. Cell Culture

Human breast cancer cell lines (HS578T, MDA-MB-468, MCF-7, MDA-MB-231, MDA-MB-436, SKBR3, and T47D) were purchased from the American Type Culture Collection (ATCC, Manassas, VA, USA). Cells were cultured in Dulbecco’s modified Eagle’s medium (DMEM; WELGENE, LM 001-05, Gyeongsan-si, Republic of Korea), supplemented with 10% fetal bovine serum (Gibco) and antibiotics (100 U/mL penicillin and 100 μg/mL streptomycin), at 37 °C in a humidified incubator with 5% CO_2_. The human normal mammary gland epithelial cell line MCF-10A was obtained from the ATCC. MCF-10A cells were cultured in DMEM/F-12 (Gibco), supplemented with 5% fetal bovine serum (Gibco), antibiotics (100 U/mL penicillin and 100 μg/mL streptomycin), 20 ng/mL recombinant human EGF (Peprotech, Rocky Hill, NJ, USA), 10 μg/mL insulin (Sigma-Aldrich, St. Louis, MO, USA), 50 ng/mL hydrocortisone (Stemcell, Vancouver, BC, Canada), and 100 ng/mL cholera toxin (Sigma-Aldrich), at 37 °C in a humidified incubator with 5% CO_2_.

### 2.2. Preparation of miRNA, DNA Constructs, and Transfection

Hsa-miR-302a, hsa-miR-302b, and hsa-miR-302c mimics were obtained from Bioneer (Daejeon, Korea). miRNA mimic sequences were as follows: hsa-miR-302a (5′-ACU UAA ACG UGG AUG UAC UUG CU-3′), hsa-miR-302b (5′-ACU UUA ACA UGG AAG UGC UUU C-3′), and hsa-miR-302c (5′-UUU AAC AUG GGG GUA CCU GCU G-3′). To generate the miRNA expression plasmid, hsa-miR302c-3p precursor primer sequences (forward: 5′-CCGGtaagtgcttccatgtttcagtggCTCGAGccactgaaacatggaag cacttaTTTTTG-3′ and reverse: 5′-AATTCAAAAAtaagtgcttccatgtttcagtggCTCGAGccactgaaacatggaag cactta-3′) were obtained from bioneer (Daejeon, Korea). Forward and reverse hsa-miR302c oligonucleotides were annealed using 5× Phusion HF buffer (New England BioLabs) and cloned into the pLKO.1-puro vector. Human ATAD2 cDNA (NM_014109.3) was purchased from Sino Biological Inc. (Beijing, China). ATAD2 cDNA was cloned into pcDNA3.1 and the pCDH-CMV-MCS-hygro vector. ATAD2 targeting shRNA plasmid DNAs were purchased from Sigma-Aldrich. All plasmid DNAs were transfected into Lenti-X 293T (Clonetech, Mountain View, CA, USA) with Mission Lentiviral Packaging Mix (Sigma-Aldrich) using Lipofectamine 3000 reagent (Thermo Scientific, Rockford, IL, USA) following the manufacturer’s instructions. Culture supernatant containing lentiviral particles was collected and filtered using a Minisart^®^ syringe filter (0.2 μm pore size, Sartorius, Goettingen, Germany). The collected lentiviral particles were infected with SKBR3 and T47D cell lines, and selected using puromycin (1 μg/mL for SKBR3 and 2 μg/mL for T47D) and hygromycin B (200 μg/mL for SKBR3 and 500 μg/mL for T47D).

### 2.3. miRNA Prediction

miRNAs regulating ATAD2 expression were predicted using TargetScanHuman 7.2 software (http://www.targetscan.org/vert_72/; accessed on 5 March 2018). The high-scoring miRNAs were used for further investigation.

### 2.4. Immunohistochemistry (IHC)

A human breast cancer tissue array (60 cases; CBA) was purchased from SuperBioChips (Seoul, Korea). The array contains 10 normal tissue sections and 50 tumor tissue sections, including stage IIA (*n* = 12), stage IIB (*n* = 8), stage IIIA (*n* = 12), stage IIIB (*n* = 4), and stage IIIC (*n* = 14). Normal and tumor tissues were deparaffinized by xylen, and antigen retrieval was carried out in citrate buffer for 10 min. The slides were washed in PBS and incubated in blocking solution. The slides were immunohistochemically stained with diluted primary antibody against ATAD2 (1:200) using a DAB substrate kit (Vector laboratories, Burlingame, CA, USA). The relative intensity of ATAD2 was quantified using open access imageJ software (http://openwetware.org/wiki/Sean_Lauber:ImageJ, accessed on 9 October 2012). The % stained area is determined as the IHC stained area (brown staining)/total area (brown + non-brown staining) × 100.

### 2.5. Reverse Transcription-Polymerase Chain Reaction (RT-PCR) and Quantitative RT-PCR (qRT-PCR)

Human breast cancer tissues and corresponding adjacent normal tissues (33 breast normal/tumor tissues pairs from the same patients) were provided by the Biobank of Chungnam National University Hospital, a member of the Korea Biobank Network. The experiments using human tumor tissues were approved by the Korea Research Institute of Bioscience and Biotechnology Review Board (P01-2016-31-002). Total RNA was extracted from breast cancer tissues using TRIzol reagent (Thermo Fisher Scientific, Waltham, MA, USA). A total of 1 μg mRNA was reverse transcribed to cDNA using the GoScript™ Reverse Transcription System (Promega, Madison, WI, USA), and 1μL of cDNA was used for RT-PCR and qRT-PCR. For RT-PCR analysis, each PCR product was electrophoresed on a 2% agarose gel. For qRT-PCR, triplicate reactions were carried out using StepOnePlus Real-Time PCR (Thermo Fisher Scientific). mRNA levels were normalized to β-actin levels, and the relative fold-change in gene expression was calculated using the 2^−ΔΔCt^ method. The primer sequences were as follows: β-actin for RT-PCR (forward: 5′-AGA AAA TCT GGC ACC ACA CC-3′ and reverse: 5′-CTC CTT AAT GTC ACG CAC GA-3′); ATAD2 for RT-PCR (forward: 5′-CCC AGA GCA GAA TGA AAA GC-3′ and reverse: 5′-TCG AGT CAT TCG CAG AAC AC-3′); β-actin for qRT-PCR (forward: 5′-CTC TTC CAG CCT TCC TTC CT-3′ and reverse: 5′-AGC ACT GTG TTG GCG TAC AG-3′); and ATAD2 for qRT-PCR (forward: 5′- ACC ACC TGA GCC AAG ATC AC-3′ and reverse: 5′-ACC TCA TCA GGG TCA ACA GG-3′).

### 2.6. Western Blotting

Cells were lysed in RIPA buffer (100 mM Tris-HCl (pH 7.4), 50 mM NaCl, 0.5% NP40, 0.5% sodium deoxycholate, 0.1 mM Na_3_VO_4_, 50 mM β-glycerophosphate, 50 mM NaF, and protease inhibitor cocktail (Sigma-Aldrich)), and proteins from cell lysates were quantified using the Pierce^®^ BCA Protein Assay Kit (Thermo). Equal amounts of protein samples (10–50 μg) were separated via 6–15% SDS-PAGE and transferred onto PVDF membranes using a Trans-Blot^®^ Turbo™ Transfer pack (Bio-Rad, Hercules, CA, USA). The membranes were blocked with 5% skimmed milk in TBS-T for 1 h and incubated overnight at 4 °C with the following primary antibodies: anti-ATAD2 (Sigma-Aldrich, HPA029424), anti-Ki67 (Santa Cruz Biotechnology, Santa Cruz, CA, USA, sc-550609), anti-Snail (Cell Signaling Technology, Danvers, MA, USA, CST-3879), anti-E-cadherin (Santa Cruz Biotechnology, sc-8426), anti-N-cadherin (Cell Signaling Technology, CST-4061), anti-Claudin (Santa Cruz Biotechnology, sc-4933), and anti-β-actin (Santa Cruz Biotechnology, sc-47778). The antibody-bound proteins were then incubated with HRP-conjugated secondary antibodies and visualized using a chemiluminescent HRP substrate (Millipore, Billerica, MA, USA). The relative intensities of the bands were measured using ImageJ.

### 2.7. Cell Proliferation Assay

Cells were plated at 1 × 10^5^ cells per well in 6-well plates. After incubation for 2 and 4 days, the cells were harvested and resuspended in 500 μL media. Viable cells were counted via the trypan blue exclusion method using a Countess II Automated Cell counter (Thermo Fisher Scientific).

### 2.8. Cell Migration and Invasion Assays

Transwell migration and invasion assays were performed with the modified Boyden chamber method using Corning^®^ Transwell^®^ polycarbonate membrane cell culture inserts (Corning, NY, USA). After transfection, 1–2 × 10^5^ cells in 100 μL serum-free medium were plated into the upper chambers. The lower chambers were filled with 600 μL DMEM containing 10% fetal bovine serum. After a 24 h (for migration) or 48 h (for invasion) incubation, cells remaining in the upper chambers were removed with a cotton swab. The migrating or invading cells were stained with crystal violet and dissolved in 10% acetic acid. Absorbance was measured at 595 nm using a microplate reader (Molecular Devices, Sunnyvale, CA, USA). The relative migratory rates were analyzed as a percentage compared to the control cells. For the invasion assay, the transwell membranes of the upper chamber were coated with Matrigel^®^ Matrix (Corning).

### 2.9. Luciferase Reporter Assay

The 3′-untranslated region (3′-UTR) luciferase reporter assay was performed to evaluate miRNA activity quantitatively. The wild-type 3′-UTR or mutant 3′-UTR of ATAD2 was introduced into the pmirGLO Dual-Luciferase miRNA Target Expression Vector (Promega), downstream of the firefly luciferase gene. These vectors were co-transfected with the miR-302 family into cancer cell lines. After 24 h of transfection, firefly and Renilla luciferase activities were quantified using the Dual-Luciferase Reporter assay system (Promega) following the manufacturer’s instructions. Luciferase activity was normalized to Renilla luciferase activity.

### 2.10. Animal Experiments

Six-week-old BALB/cAJcl-nude mice were purchased from Dae Han Bio Link Co., Ltd. (Eumseong-gun, Korea) and housed in a pathogen-free room in an animal care facility at KRIBB. The mice were randomly divided into three groups (Mock, hsa-miR-302c, and hsa-miR-302c, plus ATAD2). SKBR3 and T47D cells were suspended at a density of 2 × 10^7^ cells in 100 μL of PBS and 100 μL of Matrigel, and injected into the right flank of the mouse. Four weeks after cell injection, mice were anesthetized with avertin (500 mg/kg), and then tumor tissues were excised to measure the volume and weight. The tumor volume was calculated based on the following formula: 0.52 × (major axis) × (minor axis) × (height). Tumor tissues were lysed for Western blotting. All animal studies were performed according to the guidelines of, and with the permission of the KRIBB Institutional Animal Care and Use Committee.

### 2.11. Statistical Analysis

Experimental data were expressed as mean ± standard deviation, and all statistical analyses were performed using the Student’s *t*-test. Differences were considered statistically significant at *p* < 0.05.

## 3. Results

### 3.1. ATAD2 Expression Is Increased in Breast Cancer Tissues and Cell Lines

We first examined the mRNA expression of *ATAD2* in 33 breast normal/tumor tissue pairs via RT-PCR analysis. The mRNA expression of *ATAD2* was increased in 24 human breast tumor tissues compared to that in normal samples (Figure 1A). To further verify this result, the relative expression levels of ATAD2 were investigated via the qRT-PCR analysis of 15 paired samples of normal breast and tumor breast tissues. ATAD2 mRNA expression was approximately 4-fold higher in breast tumor tissues than in normal tissues (Figure 1B). Furthermore, in The Cancer Genome Atlas (TCGA) data, ATAD2 mRNA expression was upregulated in the human breast cancer tissues compared to the non-tumor samples (Figure 1C).

To verify the expression of ATAD2 protein in breast cancer, we performed immunohistochemistry. Intense staining for ATAD2 was observed frequently in breast cancer tissue specimens, while negative or very weak staining was observed in normal tissue specimens (Figure 1D, left panel). Quantification of staining intensity according to ImageJ software demonstrated that 14.0% of the area in the tumor stage IIA sample, 16.8% in tumor stage IIB, 29.8% in tumor stage IIIA, 33.0% in tumor stage IIIB, 44.6% in tumor stage IIIC, compared to 3.4% in normal tissues, was ATAD2 positive (Figure 1D, right panel). RT-PCR analyses showed that ATAD2 mRNA was expressed in seven breast cancer cell lines, but not in the human normal breast cell line MCF-10A. Accordingly, Western blot analysis showed that the ATAD2 protein was not detected in MCF-10A, while it was slightly expressed in two breast cancer cell lines (HS578T and MDA-MB-468) and highly expressed in five breast cancer cell lines (MCF-7, MDA-MB-231, MDA-MB-436, SKBR3, and T47D) (Figure 1E). Together, these findings show that ATAD2 expression increases in human breast cancer tissues and cell lines, suggesting that ATAD2 may be involved in tumorigenesis.

### 3.2. ATAD2 Promotes Breast Cancer Cell Proliferation, Migration, and Invasion

To investigate the possible role of ATAD2 in the malignant phenotypes of breast cancer cells, we established ATAD2-depleted SKBR3 and T47D cell lines via transfection with short hairpin RNAs (shRNAs) targeting ATAD2. The shATAD2 (shATAD2-1 and shATAD2-2) cell line showed markedly reduced expression levels of ATAD2, while the control shRNA did not influence ATAD2 expression (Figure 2A). ATAD2 depletion caused a significant reduction in the proliferation rate of each cell line (Figure 2B,C). However, there was no difference in the apoptosis of control Mock cells and ATAT2-depleted shATAD2-1 and shATAD2-2 cells (Appendix A), suggesting that the reduced proliferation by ATAD2 knockdown would not be due to apoptosis. To further investigate the effect of ATAD2 on breast cancer cell behaviors associated with malignant progression, we examined whether it affects cancer cell migration and invasion. The transwell migration assay demonstrated that ATAD2 knockdown attenuated cell motility in SKBR3 and T47D cells (shATAD2-1 and shATAD2-2) compared to that in the control shRNA-transfected cells (shcon) (Figure 2D,E). Similarly, the Matrigel invasion assay showed that the invasiveness of ATAD2-depleted SKBR3 and T47D cells was lower than that in the control cells (Figure 2D,E). These results indicate that ATAD2 promotes breast cancer cell proliferation, migration, and invasion.

### 3.3. miR-302 Family Inhibits Breast Cancer Cell Migration and Invasion by Directly Targeting ATAD2

As miRNAs regulating ATAD2 expression have been reported to suppress cancer proliferation and progression [3], we used mRNA targeting-prediction algorithms (TargetScan) to identify putative miRNAs targeting ATAD2. The miR-302 family was predicted to be a high-scoring miRNA that can target the 3′-UTR of ATAD2 (Figure 3A). To determine whether the miR-302 family can regulate ATAD2 expression, we transfected SKBR3 and T47D cells with miR-302 a, b, and c mimics or the control miRNA. Western blot data showed that ATAD2 expression was markedly reduced in cells transfected with miR-302a, b, and c mimics (Figure 3B), suggesting that miR-302 could suppress ATAD2 expression. To investigate whether miR-302 affects the proliferation of breast cancer cells, we transfected with miR-302a, b, and c mimics into SKBR3 and T47D cells. The result showed that miR-302a, b, and c mimics significantly reduced the proliferation of SKBR3 and T47D cells (Figure 3C). Next, we investigated whether miR-302 affects the motility and invasiveness of breast cancer cells. The transwell migration and invasion assay results showed that the transfection of miR-302a, b, and c mimics into SKBR3 and T47D cells significantly reduced their migration and invasion (Figure 3D,E).

To investigate whether ATAD2 is a direct target of miR-302, we generated a firefly luciferase reporter vector containing the ATAD2 3′-UTR (Figure 4A). The WT luciferase vector was transfected, along with the negative control or miR-302 a, b, or c mimics, into SKBR3 and T47D cells. Transfection with each miRNA mimic markedly reduced the luciferase activity by approximately 21–45% (Figure 4B,C). Subsequently, we generated a mutant ATAD2 luciferase construct carrying seven base pair changes in the miRNA putative binding site at the ATAD2 3′-UTR (Figure 4A). Mutation of the miRNA-binding sequences reduced the ability of miR-302 mimics to inhibit luciferase activity (Figure 4B,C), which suggested that miR-302a, b, and c directly target the 3′-UTR of ATAD2.

Next, we determined whether ATAD2 expression is involved in the inhibition of cancer cell migration and invasion by miR-302. To verify this, we transfected miR-302c mimics, along with the control empty vector or ATAD2-expressing vector, into SKBR3 and T47D cells. Western blot results showed that ATAD2 levels were restored in cells transfected with the ATAD2-expressing vector compared to those in cells transfected with the empty vector (Figure 5A). ATAD2 overexpression partially rescued the inhibition of proliferation, migration, and invasion ability caused by miR-302c in SKBR3 and T47D cells (Figure 5B–D). These results suggest that miR-302 suppresses breast cancer cell behaviors associated with malignant progression by directly targeting ATAD2.

### 3.4. miR-302 Suppresses Tumor Growth In Vivo by Targeting ATAD2

Next, we established Mock, miR-302, or miR-302/ATAD2 overexpressing stable cells from SKBR3 and T47D cells. Western blot data showed that miR-302 overexpression markedly reduced ATAD2 expression. However, ATAD2 levels were restored in cells stably transfected with the ATAD2-expressing vector (Appendix A). Restoration of ATAD2 rescued the inhibition of proliferation, migration, and invasion ability caused by miR-302c in SKBR3 and T47D cells (Appendix A). To investigate whether ATAD2 expression regulated by miR-302 is associated with tumor growth in vivo, we implanted the Mock, miR-302, or miR-302/ATAD2 expressing SKBR3 and T47D cells subcutaneously into nude mice. Mice injected with miR-302 overexpressing cells showed enhanced tumor volume and weight compared to mice injected with Mock cells. However, restoration of ATAD2 reversed the inhibitory effect of miR-302 on tumor growth (Figure 6A–C,E–G). Western blot analysis using tumor tissues showed that miR-302 overexpression downregulated the protein level of Ki-67, Snail, and N-cadherin proteins, and upregulated the level of E-cadherin and claudin. However, these effects were rescued by ATAD2 restoration (Figure 6D,H). These findings suggested that miR-302 suppressed tumor growth and the epithelial–mesenchymal transition (EMT) by inhibiting ATAD2 expression.

## 4. Discussion

ATAD2 is highly expressed in multiple cancers and plays multifaceted oncogenic functions in the proliferation and survival of cancer cells [1,2,3]. In breast cancer, ATAD2 has been identified as an ERα coregulatory factor that plays a critical role in ER-mediated activation. ATAD2 is directly associated with estrogen-bound ERα and ACTR, and thus elevates the expression of ERα-targeted cell cycle regulators, such as cyclin D1, Myc, and E2F1, which promote cancer cell proliferation [26,27]. Additionally, ATAD2 could positively mediate the expression of pro-survival genes, including SGK, VEGF, IRS2, and AKT, which contribute to promoting breast cancer progression [15]. ATAD2 expression levels are correlated with a poor prognosis in patients with breast cancer [6]. Moreover, the elevated expression of ATAD2 is further associated with tamoxifen-resistant phenotypes in breast cancer [18]. Consistent with these findings, we showed that ATAD2 is highly expressed in breast cancer tissues and cell lines, and plays a critical role in cancer cell migration, invasion, and proliferation.

miRNAs play important roles in various biological processes, including cell development, proliferation, and apoptosis, by modulating the expression of target mRNAs [28]. The abnormal expression of miRNAs can result in the development of various pathological events such as cancers. Therefore, miRNAs are considered therapeutic, diagnostic, and prognostic factors in various human cancers [29]. Recent studies have demonstrated that several miRNAs suppress cancer cell proliferation and progression. For example, miRNA-520f reduces the expression of ATAD2 to inhibit the proliferation of gastric carcinoma cells [30]. MiR-372 binds to the ATAD2 3′-UTR and downregulates its expression to inhibit the proliferation of ovarian and hepatocellular carcinoma cells [31,32]. MiR-186 further inactivates the Hedgehog signaling pathway by targeting ATAD2 in retinoblastoma cells, thereby inhibiting cell viability, migration, and angiogenesis [33]. ATAD2 has also been reported to be targeted by miR-200b-5p in ovarian cancer cells, where miR-200b overexpression suppresses proliferation and promotes the apoptosis of ovarian cancer cells by inhibiting ATAD2 expression [34].

The current study showed that miR-302 downregulates ATAD2 expression to inhibit breast cancer cell proliferation, migration, and invasion in vitro and tumor growth in vivo. Furthermore, ATAD2 overexpression rescues the inhibition of malignant phenotypes mediated by miR-302. Consistent with our results, a recent study has demonstrated that miR-302 attenuates the epithelial–mesenchymal transition and cisplatin resistance by targeting ATAD2 in ovarian cancers [35]. Furthermore, the miR-302/367 cluster has been reported to induce the mesenchymal-to-epithelial transition and suppress the proliferation and invasion of breast cancer cells [36,37]. miR-302 is downregulated in adriamycin (ADR)- and mitoxantrone (MX)-resistant MCF-7 cells. The overexpression of miR-302 sensitizes breast cancer cells to ADR and MX by targeting MEKK1 [38,39]. These findings suggest that miR-302 may function as a tumor suppressor in breast cancer.

## 5. Conclusions

In summary, we identified miR-302 as a novel miRNA that suppresses cancer cell behaviors associated with tumor progression by directly targeting ATAD2. Our data provide new insights into the mechanisms of breast cancer progression. We propose that miR-302 may be a potential therapeutic candidate for breast cancer treatment. However, the detailed mechanism of miR-302 and its targets in breast cancer development and progression remains to be further understood.

## Figures and Tables

**Figure 1 cancers-14-04345-f001:**
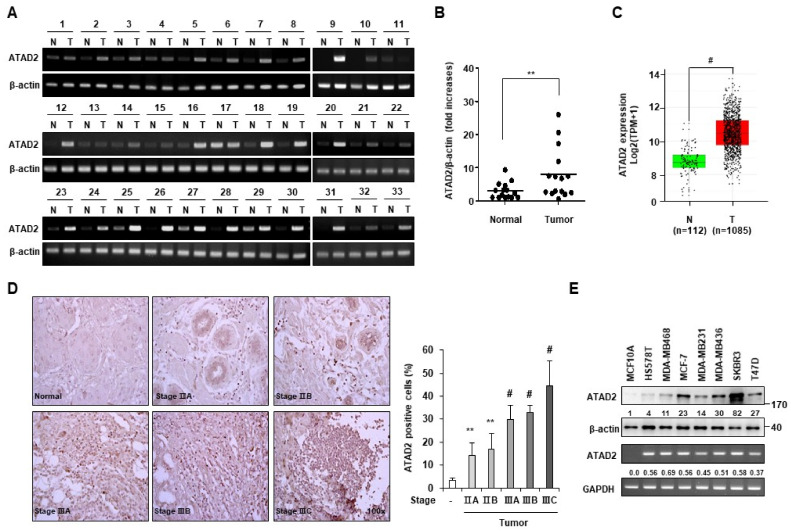
The expression of ATAD2 is elevated in breast cancer tissues and cell lines. (**A**) RT-PCR was performed to analyze the mRNA expression of ATAD2 in 33 paired samples of the non-tumor breast (N) and tumor breast (T) tissues. β-actin was used as an internal control (**B**) The relative expression level of ATAD2 mRNA was identified by qRT-PCR analysis in 15 paired samples of the non-tumor breast (N) and tumor breast (T) tissues. (**C**) ATAD2 expression in TCGA cohort using the GEPIA (http://gepia.cancer-pku.cn/detail.php, accessed on 5 September 2017). (**D**) ATAD2 expression in normal breast (*n* = 10) and breast cancer tissues including stage IIA (*n* = 12), stage IIB (*n* = 8), stage IIIA (*n* = 12), stage IIIB (*n* = 4), and stage IIIC (*n* = 14). Original magnification, 100×. The histogram shows the relative intensity of ATAD2-positive cells. The mean values and the standard error were obtained from two individual experiments. ** *p* < 0.01, # *p* < 0.001. (**E**) The level of ATAD2 protein and mRNA in human breast cancer cell lines was verified via Western blot and RT-PCR analysis, respectively. Relative intensities were measured by ImageJ. Uncropped blots can be found in Appendix A.

**Figure 2 cancers-14-04345-f002:**
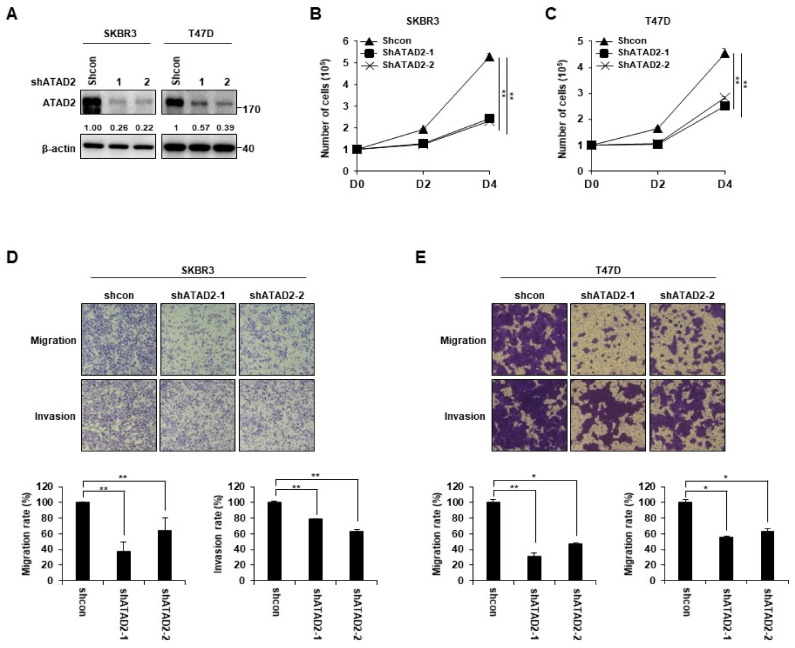
ATAD2 knockdown expression inhibits breast cancer cell proliferation, migration, and invasion. SKBR3 and T47D cells were transfected with Mock (as a control) and shATAD2 plasmid. After puromycin selection, (**A**) the protein level of ATAD2 was examined via Western blot analysis. β-actin was used as a loading control. (**B**,**C**) The proliferation of indicated cells was determined via cell counting after trypan blue staining at each time point. (**D**,**E**) Migration and invasion abilities were analyzed using the modified Boyden chamber method, as described in Materials and Methods. Graphs represent migration and invasion rates (%) according to crystal violet staining. Statistical analyses were performed using paired two-tailed Student’s *t*-test. * *p <* 0.05, ** *p <* 0.01. Uncropped blots can be found in Appendix A.

**Figure 3 cancers-14-04345-f003:**
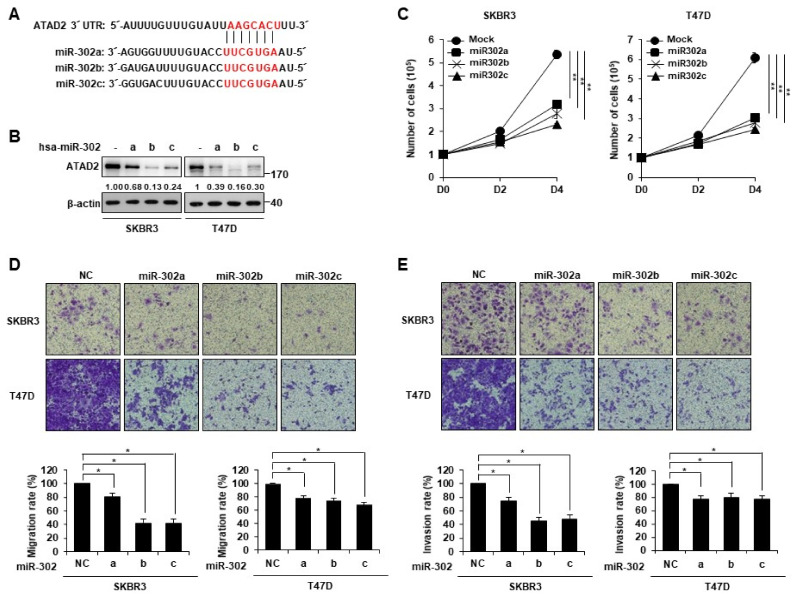
miR-302 family attenuates breast cancer cell migration and invasion by downregulation of ATAD2 expression. (**A**) The miR-302 family binding site was predicted at the 3′ UTR of ATAD2 mRNA. (**B**) SKBR3 and T47D cells were transfected with 100 nM of hsa-miR-302a, b, and c mimics. Western blot analysis was performed with the ATAD2 antibody. β-actin was used as a loading control. (**C**) The proliferation of transfected SKBR3 and T47D cells with 100 nM of negative control (NC); miR302 a, b, and c were identified via cell counting after trypan blue staining at each time point. SKBR3 and T47D cells were transfected with 100 nM of negative control (NC), miR-302a, b, and c. At 24 h after transfection, cells were seeded into the upper chamber of transwell without serum, and migration (**D**) or invasion (**E**) ability of indicated cells were measured as described in Materials and Methods. Graphs represent the relative percentages of migrating or invading cells compared to the negative control. Statistical analyses were performed using paired two-tailed Student’s *t*-test. * *p <* 0.05, ** *p <* 0.01. Uncropped blots can be found in Appendix A.

**Figure 4 cancers-14-04345-f004:**
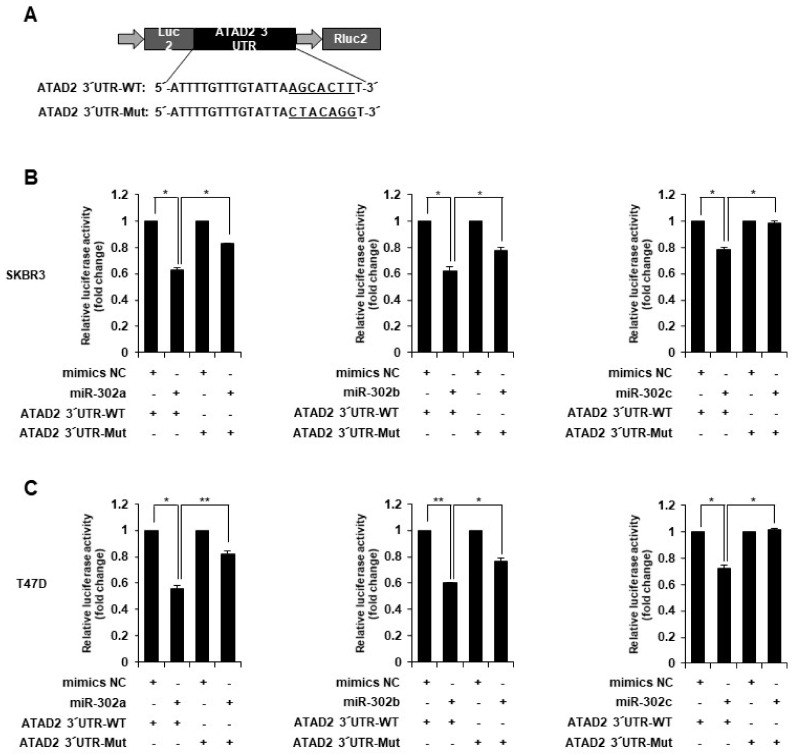
miR-302 family targets 3′-UTR of ATAD2 mRNA. (**A**) Schematic diagram of the luciferase reporter vectors containing the predicted miR-302 family binding sites of 3′-UTR of ATAD2 mRNA. WT or mutant sequences of 3′-UTR of ATAD2 mRNA were cloned into downstream region of the *Firefly luciferase* reporter gene. (**B**,**C**) WT or mutant luciferase reporter vectors were co-transfected with negative control (NC) or miR-302 family (a, b, and c) into (**B**) SKBR3 or (**C**) T47D cells. Luciferase activity was measured after 24 h transfection as described in Materials and Methods. Graphs represent relative fold-changes compared to control and show the mean of three independent experiments. Statistical analyses were performed using paired two-tailed Student’s *t*-test. * *p <* 0.05, ** *p <* 0.01.

**Figure 5 cancers-14-04345-f005:**
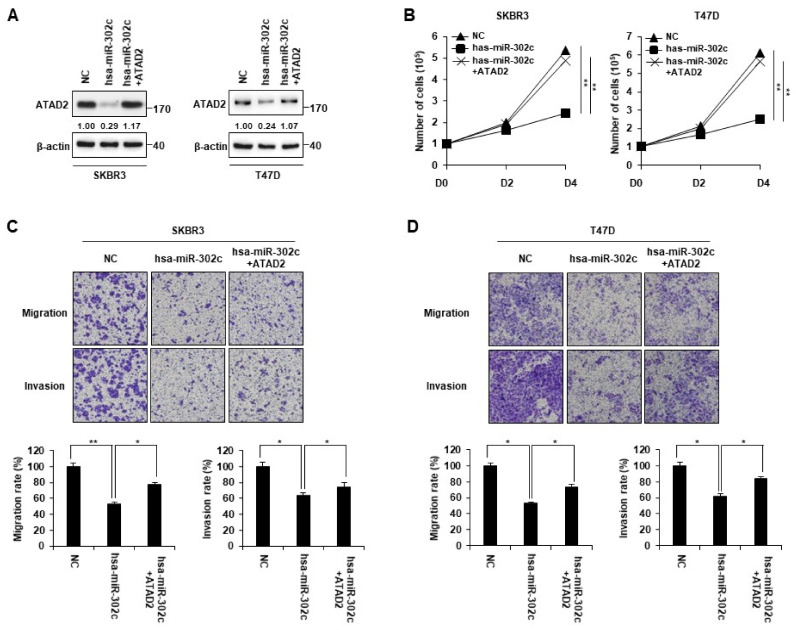
Restoration of ATAD2 expression rescues the miR-302c-mediated inhibition of breast cancer cell proliferation, migration, and invasion. hsa-miR-302c mimic was transfected with or without HA-tagged ATAD2 into SKBR3 or T47D cells. (**A**) The protein expression of ATAD2 in indicated cells was assessed via Western blot analysis. β-actin was used as a loading control. (**B**) The proliferation of SKBR3 and T47D were verified via cell counting after trypan blue staining at each time point. (**C**,**D**) hsa-miR-302c mimic was transfected with or without HA-tagged ATAD2 into (**C**) SKBR3 or (**D**) T47D cells. After 24 h transfection, transfected cells were plated into the inner chamber of the transwell without serum. Migrating and invading cells were measured using crystal violet staining as described in Materials and Methods. Graphs represent the mean of three independent experiments. Statistical analyses were performed using paired two-tailed Student’s *t*-test. * *p <* 0.05, ** *p <* 0.01. Uncropped blots can be found in Appendix A.

**Figure 6 cancers-14-04345-f006:**
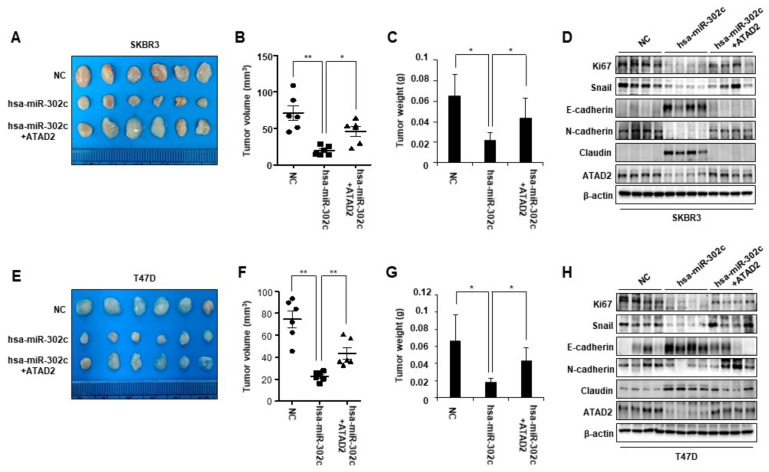
miR-302c suppresses tumor growth by targeting ATAD2. Mock, miR-302, or miR-302/ATAD2 expressing SKBR3 or T47D cells were subcutaneously injected into nude mice. (**A**,**E**) Representative photograph of tumors. Tumor volume (**B**,**F**) and tumor weight (**C**,**G**) were calculated as described in Materials and Methods. (**D**,**H**) The levels of indicated proteins in tumor tissue samples were analyzed via Western blot. Statistical analyses were performed using paired two-tailed Student’s *t*-test. * *p <* 0.05, ** *p <* 0.01.

## Data Availability

The data presented in this study are available in the Appendix A.

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
