# Peer review of "miR-302 Suppresses the Proliferation, Migration, and Invasion of Breast Cancer Cells by Downregulating ATAD2"

_cancers, 2022, doi:10.3390/cancers14184345_

Round 1
Reviewer 1 Report (New Reviewer)
This study by Hwang et al. investigates the role of ATAD2 in breast cancer and as a target for miR-302. The authors present an interesting study and use breast cancer cell lines to show that overexpression of ATAD2 in cancer tissues drives proliferation, migration, and invasion. There are a few minor concerns that need to be addressed by the authors before the manuscript could be accepted for publication.
1. Line 188: Results section 3.1.: The authors mention using “33 breast normal/tumor tissue pairs”. Is this mentioned in the materials and methods? The Source of tissues and patient information need to be included. Also, as they mention normal/tumor tissue pair, is it from the same patient? So normal means adjacent normal?
2. Line 225: Results section 3.2.: The flow cytometry data did not exhibit any significant difference in apoptotic cell percentage as observed with the Trypan Blue method. The authors are requested to provide an explanation as to why this discrepancy might have occurred.
3. Is the expression of ATAD2 similar irrespective of the type of breast cancer?
Author Response
Dear Reviewer 1
This study by Hwang et al. investigates the role of ATAD2 in breast cancer and as a target for miR-302. The authors present an interesting study and use breast cancer cell lines to show that overexpression of ATAD2 in cancer tissues drives proliferation, migration, and invasion. There are a few minor concerns that need to be addressed by the authors before the manuscript could be accepted for publication.
Response: We thank reviewer 1 for his/her careful and comprehensive evaluation of our manuscript. We have addressed the concerns suggested by the reviewer and modified the manuscript accordingly. The modified manuscript was highlighted in red. The point-by-point answers to each reviewer’s comments are described below.
Point 1. Line 188: Results section 3.1.: The authors mention using “33 breast normal/tumor tissue pairs”. Is this mentioned in the materials and methods? The Source of tissues and patient information need to be included. Also, as they mention normal/tumor tissue pair, is it from the same patient? So normal means adjacent normal?
Response: We thank the reviewer for making us aware of our error. We used human breast cancer tissues and corresponding adjacent normal tissues (33 breast normal/tumor tissue pairs from the same patients). Now we have added the information of human breast normal/tumor tissues in the Materials and Methods section (Line 136-140)
Point 2. Line 225: Results section 3.2.: The flow cytometry data did not exhibit any significant difference in apoptotic cell percentage as observed with the Trypan Blue method. The authors are requested to provide an explanation as to why this discrepancy might have occurred.
Response: We thank the reviewer for the valuable suggestion. Although ATAD2 depletion caused a significant reduction in the proliferation rate of each cell line (Figure 2B and 2C), it did not affect the apoptosis in SKBR3 and T47D cells (Supplementary Figure 1). Therefore, the reduced growth rate of cancer cells by ATAD2 knockdown would be due to the suppression of some proliferative pathways but not apoptosis. We have described the results on lines 241-244.
Point 3. Is the expression of ATAD2 similar irrespective of the type of breast cancer?
Response: Although ATAD2 expression is correlated with the breast cancer stages, there is no statistical significance with breast cancer types. Therefore, this manuscript has only described the correlation between ATAD2 expression and breast cancer stages.
Reviewer 2 Report (New Reviewer)
My dears,
Please find the comments and suggestions which I hope you will find useful and will improve your paper. Overall, I enjoyed reading it so congratulations. Best of luck in your future research.
XOXO

Author Response
Dear Reviewer 2
Please find the comments and suggestions which I hope you will find useful and will improve your paper. Overall, I enjoyed reading it so congratulations. Best of luck in your future research.
Response: We thank the reviewer 2 for his/her careful and comprehensive evaluation of our manuscript. We have addressed your concerns and modified the manuscript accordingly. The modified manuscript was highlighted in red. The point-by-point answers to each reviewer’s comments are described in an attached file. Please find the attached file.

Reviewer 3 Report (Previous Reviewer 1)
The manuscript can be accepted for publication.
Author Response
Dear Reviewer 3,
The manuscript can be accepted for publication.
Response: We thank the reviewer 3 for his/her careful evaluation of our manuscript.
This manuscript is a resubmission of an earlier submission. The following is a list of the peer review reports and author responses from that submission.
Round 1
Reviewer 1 Report
The authors have chosen a great subject and protein target to study its effect on breast cancer. Though the context of breast cancer is novel but the protein is already studied in ovarian cancer, and that study is not mentioned in the manuscript. The authors should discuss that study in the introduction and discussion. The experiments presented in the study are very preliminary. miRNA mediated regulation of ATAD2 is the strength of the manuscript. There is lot of scope to strengthen the manuscript further, since the topic is very interesting and the data look very promising.
- The authors must show the ATAD2 transcript levels in the cell lines. It will indicate if ATAD2 has any difference in protein and transcript level.
- Is there specific reason behind choosing T47D cells than MCF7 cells, inspite of lower ATAD2 expression.
- The authors should also check the TCGA database for expression analysis and overall survival analysis.
- It is very unclear from the manuscript that how ATAD2 exerts its oncogenic function in breast cancer cells.
- section 2.7. Wound healing assay is mentioned, that are missing in the manuscript.
Author Response
The authors have chosen a great subject and protein target to study its effect on breast cancer. Though the context of breast cancer is novel but the protein is already studied in ovarian cancer, and that study is not mentioned in the manuscript. The authors should discuss that study in the introduction and discussion. The experiments presented in the study are very preliminary. miRNA mediated regulation of ATAD2 is the strength of the manuscript. There is lot of scope to strengthen the manuscript further, since the topic is very interesting and the data look very promising.
Responses to the reviewer 1
We thank the reviewer 1 for his/her careful and comprehensive evaluation of our manuscript. As reviewer’s suggestions, we have discussed the study regarding ATAD2 function in ovarian cancer on line 328-329. We have revised the manuscript as indicated below to address the points raised by the reviewer 1.
1.The authors must show the ATAD2 transcript levels in the cell lines. It will indicate if ATAD2 has any difference in protein and transcript level.
Response: We thank the reviewer for this suggestion. As the reviewer’s comment, we have shown the expression level of ATAD2 mRNA in human breast cell lines (Figure 1E). We now describes the results on line 179-181.
2.Is there specific reason behind choosing T47D cells than MCF7 cells, inspite of lower ATAD2 expression.
Response: The reviewer makes an excellent point. ATAD2 expression is higher in MCF-7 cells than in T47D cells. Moreover, ATAD2 knockdown by siRNA reduced the proliferation of MCF-7 cells. However, MCF-7 cells could not move into lower chamber on our transwell migration and invasion systems. Therefore we used T47D cells for further studies.
3.The authors should also check the TCGA database for expression analysis and overall survival analysis.
Response: We thank the reviewer for this suggestion. According to the reviewer’s suggestion, we have analyzed the expression of ATAD2 and overall survival in TCGA cohort. ATAD2 expression was upregulated in the breast cancer tissues compared to the non-tumor samples. However, the correlation between ATAD2 expression and overall survival in TCGA database was not statistically significant. We now have added ATAD2 expression data in TCGA database (Figure 1C). We have describe the result on line 171-173.
4.It is very unclear from the manuscript that how ATAD2 exerts its oncogenic function in breast cancer cells.
Response: We thank the reviewer for this comment. According to the reviewer’s suggestion, we now have addressed the oncogenic function of ATAD2 in discussion section on line 292-297
5.section 2.7. Wound healing assay is mentioned, that are missing in the manuscript.
Response: We thank the reviewer for making us aware of our error. Now we have removed this sentence in Materials and Methods.
Reviewer 2 Report
The interaction occurring between miRNAs and their downstream targets is of importance in various pathological events and more importantly, the current experiment has focused on the role of miR-302 in breast cancer via targeting ATAD2. The authors have performed various techniques to confirm their hypothesis and the results are coherent and reliable. I think current manuscript will be of interest for many readers and I suggest publication of this paper. However, some changes in main text are needed to significantly improve quality of articles.
- The word targeting in title is a little confusing and I suggest authors to change it (upregulation or down-regulation is better, based on the results).
- It would be beneficial if authors just provide a statement about breast cancer in abstract and mentioning that it is most common tumor in women.
- Although authors have used some newly published articles from 2020, I suggest them to cite more new articles from 2020 and 2021 to improve quality and visibility of their work.
- Conclusion section can be elaborated and improved by adding more statements about limitations of current work and providing more directions for future studies. For instance, if novel therapeutics based on miR want to be developed, what are limtiations?
- The discussion section is short and I suggest authors to add one or two more paragraphs and extend it.
- Although authors have allocated a paragraph to discuss miRNAs in introduction, the information are not enough and need to be revised. Some mistakes also present. For instance, length of miRNAs is 19-24 nt. Furthermore, mention that miRNAs present in cytoplasm, they need to be loaded in RISC complex to obtain their function and they can be affected by upstream mediators such as lncRNAs. A newly published article can help you in this case (Doi, 10.1016/j.canlet.2021.03.025).
Author Response
The interaction occurring between miRNAs and their downstream targets is of importance in various pathological events and more importantly, the current experiment has focused on the role of miR-302 in breast cancer via targeting ATAD2. The authors have performed various techniques to confirm their hypothesis and the results are coherent and reliable. I think current manuscript will be of interest for many readers and I suggest publication of this paper. However, some changes in main text are needed to significantly improve quality of articles.
We thank the reviewer 2 for his/her careful and comprehensive evaluation of our manuscript. We have revised the manuscript as indicated below to address the points raised by the reviewer.
1.The word targeting in title is a little confusing and I suggest authors to change it (upregulation or down-regulation is better, based on the results).
Response: We thank the reviewer for valuable suggestion. We have now modified the title to eliminate reader’s confusion (targeting→downregulating).
2.It would be beneficial if authors just provide a statement about breast cancer in abstract and mentioning that it is most common tumor in women.
Response: According to reviewer’s suggestion, we addressed that breast cancer is the most common malignant tumor in women. (line 27)
3.Although authors have used some newly published articles from 2020, I suggest them to cite more new articles from 2020 and 2021 to improve quality and visibility of their work.
Response: We thank the reviewer for valuable suggestion. We have added the recent references to improve our manuscript. (ref. 19-27 and ref. 35)
4.Conclusion section can be elaborated and improved by adding more statements about limitations of current work and providing more directions for future studies. For instance, if novel therapeutics based on miR want to be developed, what are limtiations?
Response: The reviewer makes an excellent point. We now addressed the limitation in Conclusion section (line 328-329).
5.The discussion section is short and I suggest authors to add one or two more paragraphs and extend it.
Response: According to reviewer’s comment, we have now modified the discussion section to add a few sentences and one paragraph about oncogenic functions of ATAD2 and the role of miRNAs in cancer. (line 292-297 and line 302-306)
6.Although authors have allocated a paragraph to discuss miRNAs in introduction, the information are not enough and need to be revised. Some mistakes also present. For instance, length of miRNAs is 19-24 nt. Furthermore, mention that miRNAs present in cytoplasm, they need to be loaded in RISC complex to obtain their function and they can be affected by upstream mediators such as lncRNAs. A newly published article can help you in this case (Doi, 10.1016/j.canlet.2021.03.025).
Response: We thank the reviewer for valuable suggestion. As reviewer’s suggestion, we now have addressed the miRNAs information in the introduction section. (line 61-67)
Reviewer 3 Report
- All the invasion and migration quantification graphs should have representative images.
- Data point from each replicate in bar graphs should be visible to show the spread of values in the data (similar to fig 1b).
- Expression of ATAD2 is elevated in breast cancer tissues in mRNA level shown in fig 1 should be correlated to protein level by IHC/western of tumor vs normal tissue scoring
- Fig 1c should have a non-cancerous line as a control and should have quantification graph.
- Fig 2 b and c shows shRNA also affecting proliferation and not invasion and migration so the title of the study should reflect that.
- Following up on point 5, any promotion of apoptosis (and/or suppression of some proliferative pathways in cancer) should be checked following silencing of ATAD2 in vitro, since proliferation is profoundly affected.
- Proliferation should be checked and included in fig 3 and 5 in response to miR-302's and ATAD2's overexpression effects, respectively (similar to fig 2).
- The authors should test out the data in fig 3 and 5 in vivo in breast cancer flank or orthotopic breast cancer models. Flank tumor growth readouts/ tumor westerns or tissue IF (ATAD2, ki67, migration markers etc.) at endpoint can used to conclude therapeutic implications of the study.
Author Response
We thank the reviewer 3 for his/her careful and comprehensive evaluation of our manuscript. We have revised the manuscript as indicated below to address the points raised by the reviewer.
1.All the invasion and migration quantification graphs should have representative images.
Response: We thank the reviewer for valuable suggestion. As reviewer’s suggestion, we have added representative images of migration and invasion data. (Figure 2, 3, and 5)
2.Data point from each replicate in bar graphs should be visible to show the spread of values in the data (similar to fig 1b).
Response: We agree with the reviewer’s supportive opinion. As reviewer’s suggestion, we have modified relevant figures. (Figure 2-5)
3.Expression of ATAD2 is elevated in breast cancer tissues in mRNA level shown in fig 1 should be correlated to protein level by IHC/western of tumor vs normal tissue scoring
Response: Based on reviewer’s suggestion, we performed immunohistochemistry analysis to investigate the level of ATAD2 protein in normal and tumor tissues (Figure 1D). We now describes the results on line 174-179.
4.Fig 1c should have a non-cancerous line as a control and should have quantification graph.
Response: Base on reviewer’s suggestion, we have quantified the relative intensities by ImageJ in Figure 1E.
5.Fig 2 b and c shows shRNA also affecting proliferation and not invasion and migration so the title of the study should reflect that.
Response: ATAD2 depletion reduced the proliferation, migration, and invasion of breast cancer cells. We have modified the title.
6.Following up on point 5, any promotion of apoptosis (and/or suppression of some proliferative pathways in cancer) should be checked following silencing of ATAD2 in vitro, since proliferation is profoundly affected.
Response: The reviewer makes an excellent point. We conducted apoptosis analysis. The ATAD2 depletion did not affect the apoptosis in SKBR3 and T47D breast cancer cell lines (Supplementary Figure 1). Therefore, the reduced proliferation of cancer cells by ATAD2 knockdown would not be due to apoptosis. We now describes the results on line 205-206.
7.Proliferation should be checked and included in fig 3 and 5 in response to miR-302's and ATAD2's overexpression effects, respectively (similar to fig 2).
Response: We thank the reviewer for valuable suggestion. As reviewer’s suggestion, we investigated the effect of miR-302 on the breast cancer cell proliferation. miR-302a, b, and c mimics significantly reduced the proliferation of SKBR3 and T47D cells (Figure 3C). restoration of ATAD2 expression, at least partially, rescued the inhibition of proliferation caused by miR-302c in SKBR3 and T47D cells (Figure 5B). We now describes the results on line 234-236 and line 275-277.
8.The authors should test out the data in fig 3 and 5 in vivo in breast cancer flank or orthotopic breast cancer models. Flank tumor growth readouts/ tumor westerns or tissue IF (ATAD2, ki67, migration markers etc.) at endpoint can used to conclude therapeutic implications of the study.
Response: We thank the reviewer for the supportive comment. However, we were unable to conduct animal experiments on tumor growth and metastasis due to the deadline for submitting the revised manuscript.
Round 2
Reviewer 3 Report
Though the authors performed/reanalyzed additional in vitro data to satisfy my earlier concerns, work claiming to be translational cancer studies published in reputed journals such as this needs convincing in vivo data which the authors did not/could not perform due to lack of time. So my recommendation is rejection in the light of this being published in 'cancers'.